# SEARCH-GUIDED, LIGHTLY-SUPERVISED TRAINING OF STRUCTURED PREDICTION ENERGY NETWORKS

## ABSTRACT

In structured output prediction tasks, labeling ground-truth training output is often expensive. However, for many tasks, even when the true output is unknown, we can evaluate predictions using a scalar reward function, which may be easily assembled from human knowledge or non-differentiable pipelines. But searching through the entire output space to find the best output with respect to this reward function is typically intractable. In this paper, we instead use efficient truncated randomized search in this reward function to train structured prediction energy networks (SPENs), which provide efficient test-time inference using gradient-based search on a smooth, learned representation of the score landscape, and have previously yielded state-of-the-art results in structured prediction. In particular, this truncated randomized search in the reward function yields previously unknown local improvements, providing effective supervision to SPENs, avoiding their traditional need for labeled training data.

## 1 INTRODUCTION

Structured output prediction tasks are common in computer vision, natural language processing, robotics, and computational biology. The goal is to find a function from an input vector $\mathbf{x}$ to multiple coordinated output variables $\mathbf{y}$. For example, such coordination can represent constrained structures, such as natural language parse trees, foreground-background pixel maps in images, or intertwined binary labels in multi-label classification.

Structured prediction energy networks (SPENs) (Belanger & McCallum, 2016) are a type of energy-based model (LeCun et al., 2006) in which inference is done by gradient descent. SPENs learn an energy landscape $E(\mathbf{x}, \mathbf{y})$ on pairs of input $\mathbf{x}$ and structured outputs $\mathbf{y}$. In a successfully trained SPEN, an input $\mathbf{x}$ yields an energy landscape over structured outputs such that the lowest energy occurs at the target structured output $\mathbf{y}^*$. Therefore, we can infer the target output by finding the minimum of energy function $E$ conditioned on input $\mathbf{x}$: $\mathbf{y}^* = \mathrm{argmin}_{\mathbf{y}} E(\mathbf{x}, \mathbf{y})$.

In SPENs we parameterize $E(\mathbf{x}, \mathbf{y})$ with a deep neural network—providing not only great representational power over complex structures but also machinery for conveniently obtaining gradients of the energy. Crucially, this then enables inference over $\mathbf{y}$ to be performed by gradient descent on the energy function. Although this energy function is non-convex, gradient-descent inference has been shown to work well in practice, with successful applications of gradient-based inference to semantic image segmentation (Gygli et al., 2017), semantic role labeling (Belanger et al., 2017), and neural machine translation (Hoang et al., 2017) (paralleling successful training of deep neural networks with non-convex objectives).

Traditional supervised training of SPENs requires knowledge of the target structured output in order to learn the energy landscape, however such labeled examples are expensive to collect in many tasks, which suggests the use of other cheaply acquirable supervision. For example, Mann and McCallum (2010) use labeled features instead of labeled output, or Ganchev et al. (2010) use constraints on posterior distributions of output variables, however both directly add constraints as features, requiring the constraints to be decomposable and also be compatible with the underlying model's factorization to avoid intractable inference.

Alternatively, scalar reward functions are another widely used source of supervision, mostly in reinforcement learning (RL), where the environment evaluates a sequence of actions with a scalar

reward value. RL has been used for direct-loss minimization in sequence labeling, where the reward function is the task-loss between a predicted output and target output (Bahdanau et al., 2017; Maes et al., 2009), or where it is the result of evaluating a non-differentiable pipeline over the predicted output (Sharma et al., 2018). In these settings, the reward function is often non-differentiable or has low-quality continuous relaxation (or surrogate) making end-to-end training inaccurate with respect to the task-loss.

Interestingly, we can also rely on easily accessible human domain-knowledge to develop such reward functions, as one can easily express output constraints to evaluate structured outputs (e.g., predicted outputs get penalized if they violate the constraints). For example, in dependency parsing each sentence should have a verb, and thus parse outputs without a verb can be assigned a low score.

More recently, Rooshenas et al. (2018) introduce a method to use such reward functions to supervise the training of SPENs by leveraging rank-based training and SampleRank (Rohanimanesh et al., 2011). Rank-based training shapes the energy landscape such that the energy ranking of alternative $\mathbf{y}$ pairs are consistent with their score ranking from the reward function. The key question is how to sample the pairs of $\mathbf{y}$s for ranking. We don't want to train on all pairs, because we will waste energy network representational capacity on ranking many unimportant pairs irrelevant to inference; (nor could we tractably train on all pairs if we wanted to). We do, however, want to train on pairs that are in regions of output space that are misleading for gradient-based inference when it traverses the energy landscape to return the target. Previous methods have sampled pairs guided by the thus-far-learned energy function, but the flawed, preliminarily-trained energy function is a weak guide on its own. Moreover, reward functions often include many wide plateaus containing most of the sample pairs, especially at early stages of training, thus not providing any supervision signal.

In this paper we present a new method providing efficient, light-supervision of SPENs with margin-based training. We describe a new method of obtaining training pairs using a combination of the model's energy function and the reward function. In particular, at training time we run the test-time energy-gradient inference procedure to obtain the first element of the pair; then we obtain the second element using randomized search driven by the reward function to find a local true improvement over the first.

Some previous research has also used similar margin-based loss functions Peng et al. (2017); Iyyer et al. (2017), but with greedy beam search over predicated variables, unlike a SPEN, which searches the joint space via a learned representation of the energy using efficient gradient-descent inference.

Using this search-guided approach we have successfully performed lightly-supervised training of SPENs with reward functions and improved accuracy over previous state-of-art baselines.

## 2 SEARCH-GUIDED TRAINING

Search-guided training of SPENs relies on a randomized search procedure $S(\mathbf{x}, \mathbf{y}_s)$ which takes the input $\mathbf{x}$ and starting point $\mathbf{y}_s$ and returns a successor point $\mathbf{y}_n$ such that

$$R(\mathbf{x}, \mathbf{y}_n) > R(\mathbf{x}, \mathbf{y}_s) + \delta, \tag{1}$$

where $\delta > 0$ is the search margin. The choice of search margin $\delta$ is based on features of the reward function (e.g. range and plateaus) and indicates the minimum local improvement over the starting point $\mathbf{y}_s$. This also impacts the complexity of search, as smaller improvements are more accessible than larger improvements. In this work we use a simple randomized search: starting from the gradient-descent inference output, visiting variables in a random order, uniformly sampling a new state for each, and returning the new sample as soon as the reward increases more than the margin. We truncate the randomized search by bounding the number of times that it can query the reward function to evaluate structured outputs for each input $\mathbf{x}$ at every training step. As a result, the search procedure may not be able to find a local improvement, in which case we simply ignore that training example in the current training iteration. However, the next time that we visit an ignored example, the inference procedure may provide better starting points or truncated randomized search may find a local improvement. In practice we observe that, as training continues, the truncated randomized search finds local improvements for every training point.

In addition, if readily available, domain knowledge may be injected into the search to better explore the reward function, which is the target of our future work.

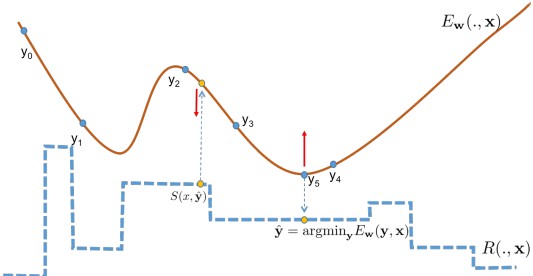

Figure 1: Search-guided training: the solid and dashed lines show a schematic landscape of energy and reward functions, respectively. The blue circles indexed by $\mathbf{y}_i$ represent the gradient-descent inference trajectory with five iterations over the energy function. Dashed arrows represent the mapping between the energy and reward functions, while the solid arrows show the direction of updates.

Intuitively, we expect that gradient-descent inference returns some $\hat{\mathbf{y}}$ as an approximate solution of $\mathrm{argmin}_y E_{\mathbf{w}}(\mathbf{x}, \mathbf{y})$. Via the search procedure, however, we find some $S(\mathbf{x}, \hat{\mathbf{y}})$ which is a better solution than $\hat{\mathbf{y}}$ with respect to the reward function. Therefore, we have to train the SPEN model such that, conditioning on $\mathbf{x}$, gradient-descent inference returns $S(\mathbf{x}, \hat{\mathbf{y}})$, thus guiding the model toward predicting a better output at each step. Figure 1 depicts an example of such a scenario.

For the gradient-descent inference to find $\hat{\mathbf{y}}_n = S(\mathbf{x}, \hat{\mathbf{y}})$, the energy of $(\mathbf{x}, \hat{\mathbf{y}}_n)$ must be lower than the energy of $(\mathbf{x}, \hat{\mathbf{y}})$ by margin $M$. We define the margin using scaled difference of their rewards:

$$M(\mathbf{x}, \hat{\mathbf{y}}, \hat{\mathbf{y}}_n)) = \alpha(R(\mathbf{x}, \hat{\mathbf{y}}_n) - R(\mathbf{x}, \hat{\mathbf{y}})), \tag{2}$$

where $\alpha > 1$ is a task-dependent scalar. Given the pairs and the margin, we can use standard margin-based training as described in Algorithm 1.

---

**Algorithm 1** Search-guided training of SPENs

---
$\mathcal{D} \leftarrow$ unlabeled mini-batch of training data
$R(.,.) \leftarrow$ reward function
$E_{\mathbf{w}}(.,.) \leftarrow$ input SPEN
**repeat**
    $\mathcal{L} \leftarrow 0$
    **for** each $\mathbf{x}$ in $\mathcal{D}$ **do**
        $\hat{\mathbf{y}} \leftarrow \mathrm{argmin}_y E_{\mathbf{w}}(\mathbf{y}, \mathbf{x})$    //using gradient-descent inference
        $\hat{\mathbf{y}}_n \leftarrow S(\mathbf{x}, \hat{\mathbf{y}})$    //search in reward function $R$ starting from $\hat{\mathbf{y}}$
        $\xi_{\mathbf{w}}(\mathbf{x}) \leftarrow M(\mathbf{x}, \hat{\mathbf{y}}, \hat{\mathbf{y}}_n) - E_{\mathbf{w}}(\mathbf{x}, \hat{\mathbf{y}}) + E_{\mathbf{w}}(\mathbf{x}, \hat{\mathbf{y}}_n)$
        $\mathcal{L} \leftarrow \mathcal{L} + \max(\xi_{\mathbf{w}}(\mathbf{x}), 0)$
    **end for**
    $\mathcal{L} \leftarrow \mathcal{L} + c||\mathbf{w}||^2$
    $\mathbf{w} \leftarrow \mathbf{w} - \lambda \nabla_{\mathbf{w}} \mathcal{L}$    //$\lambda$ is learning rate
**until** convergence

---

## 3   CITATION FIELD EXTRACTION

Citation field extraction is a structured prediction task in which the structured output is a sequence of tags such as Author, Editor, Title, and Date that distinguishes the segments of a citation text. We used the Cora citation dataset (Seymore et al., 1999) including 100 labeled examples as the validation set and another 100 labeled examples for the test set. We discard the labels of 300 examples in the training data and added to them another 700 unlabeled citation text acquired from the web.

The citation text, including the validation set, test set, and unlabeled data, have the maximum length of 118 tokens, which can be labeled with one of 13 possible tags. We fixed the length input data by padding all citation text to the maximum citation length in the dataset. We report token-level accuracy measured on non-pad tokens.

Table 1: Token-level accuracy for citation-field extraction.

| Method | Accuracy | Average reward | Inference Time (sec.) |
|---|---|---|---|
| GE | 37.3% | N/A | - |
| Iterative Beam Search (Restart=10) | | | |
| K=1 | 30.5% | -6.545 | 159 |
| K=2 | 35.7% | -4.899 | 850 |
| K=5 | 39.3% | -4.626 | 2,892 |
| K=10 | 39.0% | -4.091 | 6,654 |
| PG | | | |
| +EMA baseline | 41.8% | -13.111 | < 1 |
| +parametric baseline | 42.0% | -9.232 | < 1 |
| DVN | 29.6% | -30.303 | < 1 |
| R-SPEN | 48.3% | -9.402 | < 1 |
| SG-SPEN | **50.3**% | -10.101 | < 1 |

Our knowledge-based reward function is equivalent to Rooshenas et al. (2018), which takes input citation text and predicated tags and evaluates the consistency of the prediction with about 50 given rules describing the human domain-knowledge about citation text.

We compare SG-SPEN with R-SPEN (Rooshenas et al., 2018), iterative beam search with random initialization, deep value networks (DVNs) (Gygli et al., 2017), generalized expectation (GE) (Mann & McCallum, 2010), and recurrent neural networks trained using policy gradient methods (PG) (Williams, 1992). For PG, to reduce the variance of gradients, we used two different baseline models: exponential moving average (EMA) baseline and parametric baseline.

SG-SPEN, R-SPEN, and DVN have similar energy model, which consists of a text CNN (Kim, 2014) over joint representation of token embedding and the given tag distribution, followed by a 2-layer multi-layer perceptron.

Our reward functions of citation-field extraction dose not have access to any labeled data, and in none of our experiments the models have access to any labeled data for training or pretraining.

We reported the token-level accuracy of SG-SPEN and the other baselines in Table 1. SG-SPEN achieves highest performance in this task with 50.3% token-level accuracy. We notice that using exhaustive search through a noisy and incomplete reward function may not improve the accuracy despite finding structured outputs with higher scores.

As Table 1 indicates, the reward values for the iterative beam search is better than the reward values of both R-SPEN and SG-SPEN training methods, showing that R-SPEN and SG-SPEN training help SPENs to generalize the reward function using the unlabeled data. When the reward function is not accurate, using unlabeled data facilitates training models such as SPENs to generalize the reward function, while providing efficient test-time inference.

DVN struggles in the absence of labeled data, and having an inaccurate reward function exacerbates the situation since it tries to match the energy values with the reward values for the generated structured outputs by the gradient-descent inference. Moreover, DVN learns best if it can evaluate the reward function on relaxed continuous structured outputs, which is not available for the human-written reward function in this scenario.

## 4 CONCLUSION

We introduce SG-SPEN to enable training of SPENs using supervision provided by reward functions, including human-written functions or complex non-differentiable pipelines. The key ingredient of our training algorithm is sampling from reward function through truncated randomized search, which is used to generate informative optimization constraints. These constraints gradually guide gradient-descent inference toward finding better prediction according to reward function. We show that SG-SPEN trains models that achieve better performance compared to previous methods.

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
