# OpenReview forum: "Search-Guided, Lightly-Supervised Training of Structured Prediction Energy Networks"
_ICLR.cc/2019/Workshop/LLD — LLD 2019_

### Official Review · AnonReviewer2 · 2019-04-10
**Improving R-SPEN with margin-based loss**

**Rating:** 3
**Confidence:** 2

**Review:**

This paper studies the problem of training SPENs with indirect supervision -- human written scoring function.
Comparing to the previous work (R-SPEN), the proposed method has a simpler sampling strategy and a slightly different loss function.
Empirical performance comparison verifies the effectiveness of the proposed method.

I think for this paper, the studied problem is interesting and the proposed solution sounds novel.
My main concern is about the experiments.
The implementation details and the hyper-parameter settings are not elaborated, while the relative improvement over R-SPEN is not very significant.
Also, I think the training efficiency could be one potential advantage over R-SPEN, but this comparison is not conducted.
Overall, I think it's an interesting paper but can be further improved.

---

### Official Review · AnonReviewer1 · 2019-04-12
**Good paper on energy landscape learning**

**Rating:** 4
**Confidence:** 2

**Review:**

This article proposes a way to learn a task when no labels are available, but supposing a reward can be computed for each proposed output by the training algorithm (without limitation on the number of such computed rewards).

Instead of directly formulating this as a reinforcement learning problem, one learns an energy landscape, to fit the rewards observed, as a function of the (input, proposed output) pair; then one minimizes this energy landscape with respect to the output (by gradient descent), to find the one that will supposedly lead to the best reward for that input. This approach is known as "SPEN" (Structured Prodection Energy Networks).

To the opposite of traditional SPENs, this article suggests to combine the energy estimation with a random search; that is, during training, for a given input x, after having computed the optimal output y according to the learned energy landscape, one explores randomly around y to search for solutions leading to better rewards. This information is then used to adapt the energy landscape.
This approach is new, to my knowledge, and makes perfect sense. It also fits the workshop topic (no strongly labeled data).

Experiments on a significant task show improvement other methods.

The paper is nicely written, easy to read, even for complete beginners, despite the 4 page limitation.

---

### Decision · Program_Chairs · 2019-04-16
**Acceptance Decision**

Accept